# Biocompatible Carbon Dots/Polyurethane Composites as Potential Agents for Combating Bacterial Biofilms: N-Doped Carbon Quantum Dots/Polyurethane and Gamma Ray-Modified Graphene Quantum Dots/Polyurethane Composites

**DOI:** 10.3390/pharmaceutics16121565

**Published:** 2024-12-06

**Authors:** Zoran Marković, Sladjana Dorontić, Svetlana Jovanović, Janez Kovač, Dušan Milivojević, Dragana Marinković, Marija Mojsin, Biljana Todorović Marković

**Affiliations:** 1Vinča Institute of Nuclear Sciences, National Institute of the Republic of Serbia, University of Belgrade, Mike Petrovića Alasa 12-14, 11001 Belgrade, Serbia; sladjanaseatovic93@gmail.com (S.D.); svetlanajovanovic@vin.bg.ac.rs (S.J.); dusanm@vinca.rs (D.M.); draganaj@vin.bg.ac.rs (D.M.); 2Jozef Stefan Institute, Department of Surface Engineering, Jamova 39, SI-1000 Ljubljana, Slovenia; janez.kovac@ijs.si; 3Institute of Molecular Genetics and Genetic Engineering, University of Belgrade, Vojvode Stepe 444a, 152, 11042 Belgrade, Serbia; marija.mojsin@imgge.bg.ac.rs

**Keywords:** carbon dots, polyurethane composites, nanoelectrical properties, nanomechanical properties, antimicrobial, biocompatibility

## Abstract

**Background:** Pathogen bacteria appear and survive on various surfaces made of steel or glass. The existence of these bacteria in different forms causes significant problems in healthcare facilities and society. Therefore, the surface engineering of highly potent antimicrobial coatings is highly important in the 21st century, a period that began with a series of epidemics. **Methods**: In this study, we prepared two types of photodynamic polyurethane-based composite films encapsulated by N-doped carbon quantum dots and graphene quantum dots irradiated by gamma rays at a dose of 50 kGy, respectively. Further, we investigated their structural, optical, antibacterial, antibiofouling and biocompatibility properties. **Results:** Nanoelectrical and nanomechanical microscopy measurements revealed deviations in the structure of these quantum dots and polyurethane films. The Young’s modulus of elasticity of the carbon and graphene quantum dots was several times lower than that for single-walled carbon nanotubes (SWCNTs) with chirality (6,5). The electrical properties of the carbon and graphene quantum dots were quite similar to those of the SWCNTs (6,5). The polyurethane films with carbon quantum dots were much more elastic and smoother than the films with graphene quantum dots. Antibacterial tests indicated excellent antibacterial activities of these films against a wide range of tested bacteria, whereas the antibiofouling activities of both composite films showed the best results against the *Staphylococcus aureus* and *Escherichia coli* biofilms. Biocompatibility studies showed that neither composite film exhibited any cytotoxicity or hemolysis. **Conclusions**: Obtained results indicate that these composite films could be used as antibacterial surfaces in the healthcare facilities.

## 1. Introduction

Currently, the development of bacteria biofilms on various surfaces is a significant problem. The biofilm represents the community of pathogens embedded in a highly hydrated cellular matrix [1,2]. They can persist under very harmful conditions, such as high or low temperatures, and are resistant to antibiotics [3]. Infections triggered by biofilms are chronic. This means that they develop slowly, there is no prompt human immune system response, and ultimately, the infection can result in collateral tissue damage [4,5]. It is very difficult to treat these types of infections using drugs. The infections caused by biofilm-associated infections can be divided as follows: surface-located biofilms, which are created on biotic or abiotic surfaces, and tissue- or secretion-located biofilms. In the case of abiotic surfaces, formed biofilms serve as a precursor of intravascular infections, prosthetic joint infection, or diabetic foot ulcers [6].

In our study, we intended to produce potent antibacterial coatings based on carbon dot-encapsulated polyurethane films to eradicate various bacterial strains. Our target was plankton bacteria that appeared in the clinic or other healthcare facilities, as well as bacterial biofilms, e.g., *Pseudomonas aeruginosa*, *Staphylococcus aureus*, and *Escherichia coli*. In our previous research we encapsulated different types of carbon dots and fullerene into a polyurethane matrix and endeavored to establish the connection between antibacterial efficiency and the structure of the filler and the matrix (shape, dimension, surface chemistry, nanoelectrical and nanomechanical properties) [7,8]. Other authors have used composites of metal nanoparticles (Cu or Au) with organic dyes (i.e., methylene blue or crystal violet) encapsulated in polyurethane to achieve the antibacterial effect [9,10,11,12,13].

Carbon dots are a new class of carbon-based nanomaterials with lateral dimension lower than 100 nm [14]. This material includes the following subgroups: graphene quantum dots, carbon quantum dots, carbon nanodots, and carbon polymerized dots [15]. The methods of synthesis for the dots can be divided into two common groups, i.e., bottom-up and top-down methods. There are differences in the structures of these dots, but all types of dots exhibit unique properties, including antibacterial, anticancer, and bioimaging, for biomedical applications [16,17].

In this study, for the first time, we compared the structural, nanoelectrical and nanomechanical, optical, antibacterial, antibiofouling, cytotoxic, and hemolytic properties of two types of polyurethane-based composite films: the first one was encapsulated by N-doped carbon quantum dots (designated as CAUR-CQDs/PU), prepared using the bottom-up method, and the other was encapsulated by modified graphene quantum dots, prepared using the top-down method (designated as GQD50/PU). The gamma ray-assisted modification of graphene quantum dots (GQDs) contributes to the formation of a more defective structure, with numerous oxygen functional groups distributed over the surface and edges of a graphene-like network.

## 2. Materials and Methods

### 2.1. Materials

Citric acid (Sigma Aldrich, Rahway, NJ, USA), urea (Merck, Rahway, NJ, USA), ethanol (96 *v*/*v*%, SANI-HEM doo, Novi Bečej, Serbia), white nylon membrane filter—100 nm pore size (Tisch Scientific, Cleves, OH, USA), graphite electrodes (3 mm ⌀, purity 99.999%, Ringsdorff-Werke GmbH, Bonn, Germany), NaOH (ACS reagent, ≥97.0%, pellets, Sigma Aldrich, Burlington, MA, USA), HCl (Fisher Chemicals, Pittsburgh, PA, USA), acetone (Sigma Aldrich, Rahway, NJ, USA), SWCNT(6,5) (Sigma Aldrich, Rahway, NJ, USA), deuterated acetone (D-acetone) (Sigma Aldrich, Rahway, NJ, USA), chloroform (VWR, Avantor, Radnor, PA, USA), and deuterated chloroform (D-chloroform) (Sigma Aldrich, Rahway, NJ, USA) were purchased and used as received. Medical grade polyurethane transparent TPU film was purchased from DG Xionglin New Materials Technology, Dongguan, China. The thickness of the polymer film was 0.2 mm.

### 2.2. Synthesis and Characterization of CAUR-CQDs, GQD50 Nanoparticles, and Polyurethane-Based Composite Films

#### 2.2.1. Synthesis of CAUR-CQDs, GQD50 Nanoparticles and Corresponding Polyurethane Composite Films

N-doped carbon quantum dots (CAUR-CQDs) were prepared using the one-step solvothermal method. A mixture of 1800 mg of citric acid and 2100 mg of urea were mixed in 50 mL of ethanol. Then, the clear solution was transferred to a Teflon-lined autoclave for heating for 12 h at 180 °C. The dark red product was filtered and centrifuged. The supernatant was collected, dried, and resuspended in chloroform. A thick sediment of unreacted citric acid and urea was deposited on the bottom. After separation of the supernatant and the sediment, the supernatant was dried and resuspended in toluene. The product was again filtrated using a 100 nm nylon membrane filter.

The graphene quantum dots (GQDs) were produced using a previously described electrochemical procedure [18]; however, this procedure was modified in the GQDs isolation step. Namely, two graphite electrodes were immersed in 3 mass% NaOH in ethanol and used as the anode and cathode. The electrodes were connected to the Keithley 224 programmable current source (Cleveland, OH, USA). A 20 mA intensity current was applied, and the voltage was set at 20 V. The electrochemical reaction occurred for 8 h. During this time, the color of the electrolyte was changed from pale yellow to turbid dark red. Then, the current was stopped, and the GQDs were isolated from the electrolyte. First, the NaOH was neutralized with 10% HCl (obtained by diluting 37% *v*/*v*), after which the solvent was removed by evaporation under reduced pressure. Dried powder was dissolved in acetone. The solvent became dark, while a yellow solid residual remained. The acetone fraction was collected and characterized as graphene quantum dots (GQDs). The acetone was eliminated through evaporation. The residual powder was dispersed in cyclopentanone in a 1 mg/mL concentration. The dispersion was sonicated for 15 min using a sonication bath to achieve homogeneity. Then, the dispersion was placed in sealed glass vials and gamma-irradiated at a dose of 50 kGy using a ^60^Co gamma source in the gamma sterilization facility at the Vinča Institute. After irradiation, the solvent was evaporated under reduced pressure. The resulting residual was redispersed in acetone at a concentration of 1 mg/mL and designated as GQD50.

The CAUR-CQDs/polyurethane (CAUR-CQDs/PU) and GQD50/polyurethane (GQD50/PU) composites were prepared as follows: pieces of the polyurethane samples (25 × 25 × 1 mm^3^) were dipped in CAUR-CQDs/GQD50 toluene/acetone solution (50 mL), respectively. To encapsulate CAUR-CQDs/GQD50 into a polyurethane polymer matrix, the swelling–encapsulation–shrink method was used [8,19]. The swelling–shrinking procedure lasted 12 h/1 h at room temperature, respectively. The CAUR-CQDs/GQD50 thin foils were dried at 80 °C for 12 h in a vacuum furnace to eliminate trapped organic solvents.

#### 2.2.2. Characterization of CAUR-CQDs, GQD50 Nanoparticles, and Corresponding Polyurethane Composite Films

Surface visualization of CAUR-CQDs and GQD50 nanoparticles and their polymer composite films was performed using an atomic force microscope (AFM) with an AC160 cantilever (MFP-3D Origin, Asylum Research, Oxford Instruments, Santa Barbara, CA, USA). The particle sizes (diameters and heights) of the investigated nanoparticles was calculated using Gwyddion software 2.64 [20]. Samples for the AFM measurements were prepared as follows: Freshly cleaved mica was glued on to AFM specimen disks (Ø15 mm, Ted Pella, Redding, CA, USA). Using a spin coater, highly diluted solutions of CAUR-CQDs/GQD50 were deposited on a mica substrate. Then, the mica was annealed in a vacuum furnace for 12 h at 70 °C to remove any residual solvent. After that step, the mica was used for the AFM measurements of the nanoparticles. Polymer composites were glued on to AFM specimen disks and immediately used for AFM measurements.

As the reference for nanomechanical and nanoelectrical analysis, single-walled carbon nanotubes (SWCNTs (6,5)) functionalized with sodium deoxycholate were used. The reference sample was prepared as follows: 5 mg of SWCNT (6,5) and 0.31 g of sodium deoxycholate were mixed in 15 mL of distilled water. Then, a drop of sodium hydroxide (5M) was added. The prepared solution was treated with ultrasound for 30 min and centrifuged for 30 min at 3500 rpm. The supernatant was filtered through a nylon membrane filter with pore sizes of 450 nm and then 200 nm using a vacuum pump. The filtrate was deposited on mica using the same method as that used for the CAUR-CQDs/GQD50 colloids. 

The nanoelectrical measurements (charge distribution in polymer composite films) were conducted using the same microscope with a Si cantilever coated with Ti-Ir (ASYELEC-01, ASYELEC-01, Asylum, Oxford Instruments, Santa Barbara, CA, USA). For the measurements, we used a tip voltage of 3 V and a drive amplitude of 50 mV. As a support disc, we used an AFM/SPM Stainless Steel Metal Specimen Support Disc (Ø15 mm, Ted Pella, USA) to which the mica and polymer composite films were attached.

The nanomechanical measurements were conducted using the same microscope operated in the amplitude modulation–frequency modulation viscoelastic mapping mode using the same cantilever used previously [8]. To determine the distribution of the Young’s modulus of elasticity, frequency, amplitude, and phase of the two modes, a contact mechanics model was used [21].

The chemical composition of all samples was determined using the following two methods: X-ray photoelectron spectroscopy (XPS) and Fourier transform infrared spectroscopy (FTIR). The XPS measurements of the CAUR-CQDs and GQD50 nanoparticles were conducted on a PHI-TFA XPS spectrometer manufactured by Physical Electronics Inc. The surface composition was quantified by means of XPS peak intensities [22]. All data were calculated using Multipak software, version 9.9 [23].

The other method used for determining the chemical composition of the nanoparticles and composite film samples was the FTIR technique. All sample measurements were performed on a Nicolet iN10 Thermo Fischer Scientific device. The device operated in ATR mode, and the FTIR spectra measurements were performed under ambient conditions in the range of 400 to 4000 cm^−1^.

The photoluminescence measurements of both the CAUR-CQDs and GQD50 nanoparticle samples, as well as the composite films samples, were performed using a Fluoro-max+4 spectrofluorometer, Horiba, Kyoto, Japan under ambient conditions. The excitation wavelengths were in the range of 325 to 550 nm.

### 2.3. Reactive Oxygen Production

The detection of the singlet oxygen production of the nanoparticles and composite film samples was performed using two techniques: the first method is based on the usage of Singlet Oxygen Sensor Green (SOSG) as a fluorescence probe, and the other is based on the direct measurements of singlet oxygen luminescence at 1280 nm [24].

The singlet oxygen generation of the CAUR-CQDs and GQD50 nanoparticles was conducted as follows: the luminescence of singlet oxygen was measured at 1280 nm directly on a FluoroMax+4 fluorimeter (Horiba, Kyoto, Japan). For the luminescence measurements, we used 900 grooves/mm emission blazed monochromator gratings at 1500 nm for NIR range. As the light excitation source, an ozone-free 150 W xenon lamp was used. The emission spectra ranged from 1200–1300 nm using a thermoelectrically cooled InGaAs detector.

The samples for the direct measurement of the luminescence of singlet oxygen at 1280 nm were prepared in the following way: dry samples of CAUR-CQDs/GQD50 nanoparticles were dissolved in deuterated chloroform/acetone, respectively. The absorption of the solutions at 367 nm was tuned to 0.1 using a UV–VIS instrument. Then, the samples were excited at 367 nm, and the luminescence of singlet oxygen was detected.

The potential for singlet oxygen generation of the polymer composite films samples was evaluated as follows: the SOSG concentration in a methanol solution was 12 µmol. The composite films samples (8 × 8 mm^2^) were immersed in the SOSG methanol solution. Before measurement, the composite samples were irradiated using a blue lamp (3 W, V-TAC, Sofia, Bulgaria) from 1 to 120 min. The lamp wavelength was 470 nm. Immediately after blue light irradiation, the measurement of the luminescence of singlet oxygen was performed on a Fluoromax+4 spectrofluorometer (Horiba, Kyoto, Japan). The PL spectra of the composite films samples were measured under a 488 nm excitation in the range of 500–800 nm. The measurement step was 1 nm.

### 2.4. Antibacterial Activity of Polyurethane Composite Films

The antibacterial activities of both types of composite films were established by using ISO 22196 (plastics—measurement of antibacterial activity on plastic surfaces) [25]. Eight bacterial strains were tested to check the antibacterial potentials of these composite films: *Staphylococcus aureus* NCTC 6571 (*S. aureus*), *S. aureus* MRSA ATCC 43300 (MRSA), *Enterococcus faecalis* ATCC 29212 (*E. faecalis*), *Pseudomonas aeruginosa* ATCC 10332 (*P. aeruginosa*), *Klebsiella pneumonie* ATCC BAA2146 (*K. pneumonie*), *Listeria monocytogenes* NCTC 11994 (*L. monocytogenes*), *Escherichia coli* NCTC 9001 (*E. coli*), and *Acinetobacter baumanii* ATCC 19606 (*A. baumanii*). The composite films samples were inoculated with a 0.2 mL bacteria suspension of 1–5 × 10^8^ cell/mL. The dimensions of the composite films samples were 2.5 × 2.5 cm^2^, and they were sterilized using a UV lamp at 258 nm for 30 min. The test inoculums were covered with films, and the Petri dishes holding the test specimens were incubated for 24 h at 35 °C; humidity 90%. The bacteria were recovered with 10 mL of SCDLP, and 10 µL (different dilutions) was plated on LB agar and incubated for 24 h at 35 °C, after which the bacterial colonies were counted. Non-irradiated specimens were used for the control of bacterial viability, and other specimens were incubated under blue light at 470 nm for 1 h. A blue lamp was placed 50 cm from the surface of the samples before the measurements were performed.

The antibacterial activity of the investigated material was determined as follows [7,8]:

N—number of viable bacteria recovered for the test specimen.
N = (100 × C × D × V)/A(1)

C—average plate count; D—dilution factor for the plate counted; V—volume (mL) of SCDLP added; A—surface area (mm^2^) of the cover film.

R—antibacterial activity of the tested material.
R = (U_t_ − U_0_) − (A_t_ − U_0_) = U_t_ − A_t_(2)

U_0_—average of log10 of the number of bacteria from the non-irradiated sample in 0 h; U_t_—average log 10 of the number of bacteria from the non-irradiated sample after 24 h; A_t_—average log 10 of the irradiated sample after 24 h.

### 2.5. Antibiofouling Activity of Polyurethane Composite Films

Three bacteria biofilms—*Staphylococcus aureus* NCTC 6571, *Pseudomonas aeruginosa* ATCC 10332 and *Escherichia coli* NCTC 9001—were used to test the antibiofouling activities of the CAUR-CQDs/PU and GQD50/PU composite films. Details of the experimental procedure can be found in our previous studies [7,8].

### 2.6. Biocompatibility Studies of Polyurethane Composite Films

#### 2.6.1. Cell Culture

HaCaT cells (human immortalized keratinocytes) were purchased from the ATCC (American Type Culture Collection, Manassas, VA, USA). The cells were grown in Dulbecco’s Modified Eagle Medium High Glucose, supplemented with 10% Fetal Bovine Serum and Antibiotic-Antimycotic solution, 10.000 units/mL of penicillin, 10.000 µg/mL of streptomycin, and 25 µg/mL of Amphotericin B, all purchased from Thermo Fisher Scientific (Waltham, MA USA). The cells were cultured under optimal growing conditions, 37 °C, 5% CO_2_, and a humidified atmosphere, in a cell incubator.

#### 2.6.2. Cell Viability Assay

To evaluate the cytotoxic effect of the CAUR-CQDs/PU and GQD50/PU composite films, each was incubated in cell growth medium in a concentration 0.1 g/mL for 24 h at 37 °C. The obtained medium was considered a 100% incubation medium, and it was further diluted in the cell culturing medium to the final concentrations of 1, 10, 25, 50, and 75%.

The cytotoxicity was measured using a cell viability MTT assay [26]. It is a colorimetric assay that detects the conversion of the MTT agent (3-[4,5-dimethylthiazol-2-yl]-2,5 diphenyl tetrazolium bromide) to formazan in viable cells with an active metabolism. In brief, HaCaT cells were seeded in 96-well plates (1 × 10^5^ cells per well) and left overnight to allow the cells to attach to the surface. The next day, the cells were treated with the prepared dilutions of the incubation media. After 24 h of treatment, the MTT dissolved in the cell culturing medium, and a final concentration of 0.5 mg/mL was added to each well. After 2 h of incubation, the medium was discarded, and the formazan crystals formed in the live cells were dissolved in Dimethyl sulfoxide (DMSO, Sigma Aldrich, Rahway, NJ, USA). The absorbances were measured on a TECAN Infinite 200 PRO (TECAN, Männedorf, Switzerland) device at 560 nm.

#### 2.6.3. Hemolysis Assay

To test the potential of the CAUR-CQDs/PU and GQD50/PU composite films to mediate the hemolysis of human red blood cells (RBC), a hemolysis assay was performed [27]. Blood samples (2.5 mL) were collected from healthy human volunteers, according to the requirements of the Ethics Committee of the Institute of Molecular Genetics and Genetic Engineering, University of Belgrade (Ethical Permission No: O-EO-044/2023). Written informed consent was obtained from all human participants engaged. Fresh blood, mixed in a 9:1 ratio with 3.8% citrate as an anticoagulant, was centrifuged at 1500× *g* for 15 min to pallet the red blood cells (RBC). After discarding the supernatant, the RBC pallet was washed five times in phosphate buffer saline (PBS) and diluted in fresh PBS to a final volume of 25 mL. A total of 250 µL of this RBC dilution was used for a single reaction in the hemolysis assay. The tested materials were incubated in 750 µL PBS (0.1 g/mL) at 37 °C for 30 min, then 250 µL of dissolved RBC was added to the reaction, and incubation continued for an additional 1 h. Dissolved RBC, incubated in 750 µL of PBS, in the absence of tested materials, was used as a negative control. Dissolved RBC, incubated in 750 µL of deionized water, which induced complete lysis of RBC in hypotonic conditions, was used as the positive control. After incubation, the samples were centrifuged at 10,000× *g* for 5 min, and 100 µL aliquots of supernatant were transferred to the wells of the 96-well plate. The absorbance of the released hemoglobin was recorded on the Infinite 200 PRO microplate reader (Tecan, Männedorf, Switzerland) at 570 nm. The hemolysis rate was calculated using the following equation [28]:$$\mathrm{Hemolysis}\;\mathrm{ratio}\;\left(\%\right)=\frac{\mathrm{ODsample}-\mathrm{ODnegative}\;\mathrm{control}}{\mathrm{ODpositive}\;\mathrm{control}-\mathrm{ODnegative}\;\mathrm{control}}\times 100\%$$

## 3. Results

### 3.1. Surface Morphology of CAUR-CQDs and GQD50 Nanoparticles

To study the surface morphology of the CAUR-CQDs and GQD50 nanoparticles, we used AFM to visualize their shape and size. Appendix A show the shape, particle size distribution, and height profile of the CAUR-CQDs and GQD50 nanoparticles. Based on the data presented in Appendix A, we established that more than 95% of the CAUR-CQDs exhibited a diameter of 4.2 nm, a quasi-spherical shape, and a height of 0.4 nm. In the case of the GQD50 nanoparticles, they display a disc-like shape, and more than 90% of the nanoparticles exhibit a diameter of 40 nm and a height of 0.25 nm. Statistical analysis of the diameters and heights was performed using Gwyddion 2.64 software on more than 20 images [20].

Figure 1a–f presents the EFM and AMFM (viscoelastic measurements) for the CAUR-CQDs nanoparticles. A sample of SWCNT (6,5) was used as reference material to compare charge distribution and Young’s modulus of elasticity to the those of the CAUR-CQDs and GQD50 nanoparticles. The band gap of SWCNT(6,5) is 1.07 eV [29].

From Figure 1b,c, we noticed that the charge content and the distribution of the CAUR-CQDs nanoparticles depended on the particle diameter, i.e., for smaller nanoparticles (diameter to 4.2 nm—number 2 in Figure 1c), the charge carriers are electrons (negative well-depth of 0.6 degrees) and the distribution is nearly homogenous. But for larger nanoparticles (diameter higher than 10 nm—number 1 in Figure 1b), two opposite charges were detected: one positive (height of 1 degree) and one negative (negative well-depth of 2 degrees). The presence of an intensive positive charge can be explained by the presence of C–O bonds that are polarized towards oxygen. For smaller nanoparticles, the number of C–O bonds is too small to be detected by EFM. Since the well-depth for larger nanoparticles is nearly identical to the value of SWCNT (6,5) (Appendix A), we assumed that the CAUR-CQDs were also semiconducting. Appendix A shows the EFM images of SWCNT (6,5).

Figure 1d,e shows the top view of the AMFM image of the CAUR-CQDs (height retrace mode) and the corresponding image of the Young’s modulus of elasticity (Young’s retrace mode). Figure 1f presents the profile of the Young’s modulus of elasticity, and we determined that the Young’s modulus of elasticity is 1.15 GPa. Appendix A shows the AMFM images of SWCNT. The reference sample material SWCNT (6,5) has a Young’s modulus of elasticity in the range of 6–12 GPa (Appendix A), whereas the theoretical value is 1.048 TPa [30]. The nanomechanical measurements indicated that the Young’s modulus of elasticity of CAUR-CQDs is 5–10 times lower than that for SWCNT (6,5). The content of sp^2^ bonds in the CAUR-CQDs is much lower than that in SWCNT (6,5). The structure of CAUR-CQDs is dominated by small-diameter islands of sp^2^-connected carbon.

Khoei et al. showed theoretically that the presence of oxygen functional groups on the surface and edges of the basal plane of graphene oxide contributes to the reduction of the Young’s modulus of elasticity of graphene oxide compared to that of graphene [31]. Moreover, the increase in the oxygen functional groups contributes to increasing the C–C bond length at each hexagonal lattice of the basal plane of the CAUR-CQDs.

Figure 2a–f presents the EFM and AMFM measurements of GQD50. These dots differ from those of the CQDs due to their more crystalline structure and almost graphene-like basal plane. Figure 2a,b shows the top view AFM image and the corresponding EFM (nap retrace mode) of the GQD50 nanoparticles.

All nanoparticles in Figure 2b display white and black regions in their structure that indicate two opposite charges. In Figure 2c, the nanoelectrical profile for a single GQD50 is presented, including both positive (height 3 degree) and negative (negative well depth 2 degrees). The intensive positive charge due to the C–O bonds is three times higher than in the case of the CAUR-CQDs. The EFM phase well-depth in Figure 2c is nearly identical to the value of SWCNT (6,5) and the CAUR-CQDs. Therefore, we assumed that GQD50 was also semiconducting.

Figure 2d,e shows the top view AFM image of GQD50 (height retrace mode) and the corresponding AMFM (Young’s retrace mode). The Young’s modulus of elasticity was determined from the profile of a single GQD50 nanoparticle (Figure 2f). From this figure, we noticed that a single GQD50 had a Young’s modulus of elasticity of 0.37 GPa. This value is three times lower compared to that of the CAUR-CQDs nanoparticle and almost two orders of magnitude lower than that for SWCNT (6,5). Due to the gamma irradiation effects, the functionalization of the GQDs is intensive. The charge mobility in GQD50 is highly determined by a large number of defects. The conductance of the graphene-based Schottky diode decreases according to the gamma irradiation dose [32].

### 3.2. Surface Morphology of CAUR-CQDs/PU and GQD50/PU Composite Films

Surface morphology, including the EFM and AMFM (viscoelastic measurements) of CAUR-CQDs/PU and GQD50/PU composite films, is shown in Figure 3 and Figure 4. Figure 3a presents the morphology of the CAUR-CQDs/PU composite films (height retrace mode), and Figure 3b shows the EFM image (nap retrace mode), i.e., the charge distribution inside the polymer matrix. Black voids inside the red circle represent the CAUR-CQDs nanoparticles distributed inside the polymer pores. The size of these black voids indicates that carbon dots have formed clusters inside the polymer matrix. From Figure 3b, we concluded that there was no homogeneous distribution of the CAUR-CQDs nanoparticles inside the polymer matrix. The root-mean-square (RMS) roughness of this polymer composite is 1.93 nm compared to the RMS of 3.14. nm for neat PU film. After the encapsulation of the CAUR-CQDs nanoparticles into the polymer films, the RMS decreased 1.63 times, and in this way, the polymer surface becomes smoother. Figure 3c,d shows the surface morphology and AMFM images (Young’s retrace mode) of the CAUR-CQDs/PU composite films.

The EFM image of the GQD50/PU composite films (Figure 4b) shows the charge distribution in the interior of the GQD50/PU composite. Black voids inside the red circles represent the GQD50 inside the polyurethane polymers, which formed clusters. This figure indicates an inhomogeneous distribution of GQD50 inside the polymer matrix. Figure 4a presents the surface morphology of this composite film. The RMS of this composite film is 9.35 nm, which is almost five times higher than that of the CAUR-CQDs/PU composite films. This result indicates that the surface of the composite films becomes rougher. In Figure 4c,d we visualized the surface morphology and viscoelastic mode of the GQD50/PU composite films.

### 3.3. Chemical Composition

XPS and FTIR techniques were used to determine the elemental composition and the characteristic bonds. The FTIR spectra of the CAUR-CQDs (curve 1) and GQD50 (curve 2) nanoparticles are presented in Appendix A. In the CAUR-CQDs sample, O–H stretching vibrations were detected at 3254 cm^−1^, whereas C–H stretching vibrations were identified at 2962, 2929, and 2866 cm^−1^. The peak at 1662 cm^−1^ represents C=N stretching vibrations, whereas the peak at 1632 cm^−1^ could be assigned to C=C. C–H bending vibrations were detected at 1446 and 759 cm^−1^, while O–H bending vibrations were identified at 1382 and 1359 cm^−1^. C–O stretching vibrations were detected at 1256, 1217, and 1168 cm^−1^ [33].

The GQD50 nanoparticle sample displays the following peaks in the FTIR spectrum: peaks at 3364 and 2624 cm^−1^ due to O–H stretching vibrations, a peak at 3015 cm^−1^ stemming from C–H stretching vibrations; a peak at 1761 cm^−1^ could be assigned to C=O bonds, while a peak at 1581 cm^−1^ was due to C=C stretching vibrations; O–H bending vibrations were detected at 1391 cm^−1^, and C–O stretching vibrations were identified at 1059 cm^−1^.

Appendix A presents the FTIR spectra of the neat PU (black curve), CAUR-CQDs/PU (red curve), and GQD50/PU (green curve) composite films. The detected peaks and related bonds of neat PU were described previously in Ref. [8]. In the CAUR-CQDs composite films sample, we detected the peaks related to O–H (3326 cm^−1^) and C–H stretching vibrations (2936, 2855 cm^−1^); C=O (1726 and 1703 cm^−1^) and C=C (1598 and 1531 cm^−1^); O–H bending vibrations (1409, 1370, 1309 cm^−1^); C–O (1223, 1109, 1071 cm^−1^) and C–H bending vibrations (814, 762 cm^−1^). The peak at 1662 cm^−1^, which results from the C=N bonds, could not be detected due to overlapping with the C=C and C=O bands. The same peaks at the same positions were identified for the GQD50/PU composite film sample.

To further investigate the chemical composition of the CAUR-CQDs and GQD50 nanoparticles, we conducted XPS measurements. Appendix A shows the survey spectra of the CAUR-CQDs and GQD50 nanoparticles. Table 1 and Table 2 list the content of the elements and % of bonds detected in these two samples. Based on data from Table 1, we noticed that the content of C was 83 at%, O was 9.3 at%, and N was 7.7 at% in the CAUR-CQDs sample, whereas the content of the elements in the GQD50 sample was the following: C (83.2 at%), O (16.4 at%), and N (0.4 at%). Figure 5 and Appendix A show the XPS deconvoluted spectra of the C1s and O1s peaks of the CAUR-CQDs and GQD50 nanoparticles, whereas Appendix A shows the deconvoluted N1s peak of the CAUR-CQDS nanoparticles. The C1s peak of the CAUR-CQDs nanoparticles was fitted to three peaks at ca. 284.8 eV, ca. 286.3 eV, and ca. 288.9 eV, which corresponded to the C–C (C1), C–N (C2), and C=O (C3) bonds, respectively. The O1s peak of this sample was deconvoluted into the following three peaks: the peak at ca. 531.1 eV, the peak at ca. 532.3 eV, and the peak at ca. 533.6 eV, which corresponded to O=C–N, O=C, and O–C, respectively. The N1s peak of this sample was fitted to two peaks: one at ca. 399.6 eV and the other at ca. 398.4 eV, which corresponded to the pyridine bonds and the C=N–C bonds, respectively.

The C1s peak of the GQD50 sample was deconvoluted to three peaks at ca. 284.8 eV, 286.1 eV, and 288.4 eV, which could be assigned to the C–C (C1), C–N (C2), and C=O (C3) bonds, respectively. The O1s peak was fitted to three peaks as well: one at ca. 531.1 eV, 532.4 eV, and 533.5 eV, which corresponded to C=O (aromatic), O=C (aliphatic), and O–C, respectively.

### 3.4. Photoluminescence of CAUR-CQDs and GQD50 Nanoparticles and Polymer Composite Films

Photoluminescence is one of the very important properties of carbon dots which occurs due to their intrinsic properties, as well as their surface states. In this way, PL can be controlled by tuning the inner core and surface functional groups [34]. Carbon dots emit light in different parts of the electromagnetic spectra, e.g., from blue to red, depending on the surface chemistry of certain dots, such as their functional groups (oxygen- or nitrogen-based functional groups). In our study, we investigated the PL emission spectra of CAUR-CQDs, GQD50 nanoparticles, and CAUR-CQDs/PU and GQD50 composite films (Appendix A). This figure shows the excitation–emission dependence of the PL spectra of all samples. For the CAUR-CQDs nanoparticles, the highest emission PL intensity was obtained for the excitation wavelength of 375 nm (Appendix A). The red shifts occurred from 27–94 cm^−1^. After the CAUR-CQDs nanoparticle encapsulation in the polymer matrix, there are red shifts between the emission and excitation wavelengths (37–125 cm^−1^).

The fitting procedure for the PL emission of the CAUR-CQDs and GQD50 nanoparticles (Appendix A) reveals that the core contribution to PL of CAUR-CQDs is much greater than for GQD50. This fact can be related to the lower content of the sp^2^ bonds in defective GQD50 than in CAUR CQD, as disclosed earlier by the nanomechanical measurements [35].

### 3.5. ROS Production

One of the possible antibacterial mechanism pathways of photoactive nanoparticles is to damage the bacterial membrane wall via the production of ROS [36]. Thus, it is of the utmost interest to establish whether or not the investigated nanoparticles produce reactive oxygen species, i.e., singlet oxygen. In our work, we studied the production of singlet oxygen using two methods: by direct measurement of the intensity of the luminescence of singlet oxygen at 1280 nm and by the measurement of the PL of SOSG used as a fluorescence probe at 530 nm. Figure 6a–d shows the singlet oxygen generation of the CAUR-CQDs and GQD50 nanoparticles and the CAUR-CQDs/PU and GQD50 composite films, respectively.

The other method of measurement of singlet oxygen production showed that both the samples and the composite films produced singlet oxygen. Thus, the carbon dots encapsulated into the polymer matrix retain the ability to produce singlet oxygen. This is possible due to the porous structure of the polymer films, and we found that singlet oxygen diffused through the pores in the polymer matrix in the environment employed in our previous research [37]. The quantity of singlet oxygen is slightly larger for the GQD50/PU than for the CAUR-CQDs/PU composite films.

In this study we tested the composite films samples for the production of hydroxyl radicals using the measurement of the PL of hydroxyterephthalic acid in which the composite films samples were dipped. For either sample, there was no hydroxyl radical production under any conditions (with or without blue light irradiation). In this way, we concluded that the main mechanism of singlet oxygen production is the energy transfer from the photosensitizers (CAUR-CQDs and GQD50 nanoparticles) to molecular oxygen [38].

### 3.6. Antibacterial Activity of CAUR-CQDs/PU and GQD50/PU Composite Films

Since the investigated composite films produce singlet oxygen under blue light irradiation, these films hold the significant possibility of deactivating various bacterial strains. Surface charge, wettability and surface roughness of investigated samples have significant contributions to bacterial death [39]. Appendix A and Table 3 list the number of viable bacteria recovered per cm^2^ per test sample and the antibacterial activity of the CAUR-CQDs/PU and GQD50/PU composite films against eight bacterial strains.

The CAUR-CQDs/PU composite films demonstrate strong to moderate antibacterial potential against a wide range of tested bacteria, except for *P. aeruginosa*. The GQD50/PU composite films exhibit strong antibacterial potential against *K. pneumonie* and *E. faecalis* and cause a decrease in viable bacteria in *L. monocytogenes*.

The adhesion of bacterial strains on the coating’s surface differs depending on the type of bacterial strains tested. Namely, Gram-negative bacteria (*E. coli*) are negatively charged due to the presence of lipopolysaccharides, and their membrane wall is thinner. On the contrary, Gram-positive bacteria have a more rigid membrane wall [40]. In our study, the CAUR-CQDs/PU composite films acted bactericidally against Gram-positive (*S. aureus*, MRSA, *E. faecalis*) and Gram-negative bacteria (*K. pneumonie*), whereas the GQD50/PU composite films acted bactericidally against both Gram-positive and Gram-negative bacterial strains (only on *E. faecalis*, *K. pneumonie* and *L. mitocytogenes*). The superior antibacterial activity of CAUR-CQDs/PU composite films could be explained by the higher concentration of CAUR-CQDs nanoparticles in the polyurethane polymer matrix due to the 10-times-lower diameter of the CAUR-CQDs nanoparticles compared to those of GQD50.

The possible antibacterial mechanism of these composite films against various bacterial strains is based on the entrance of singlet oxygen through the pores of the bacterial cell membranes in the interior of the bacteria and further intracellular action [41,42]. The CAUR-CQDs and GQD50 nanoparticles produced singlet oxygen, which diffused through the pores of the polymer matrix and eradicated the bacterial strains present on the surface of the polymer.

### 3.7. Antibiofouling Activity of CAUR-CQDs/PU and GQD50/PU Composite Films

Apart from the fact that biofilms are key factors in microscale ecosystems such as subbarrier or marine biofilms [43], the significant presence of biofilms can be found in healthcare facilities on various surfaces such as door handles, switches, keyboards, bed rails, etc., including hospital instruments, triggering different infections. Figure 7 shows the antibiofouling activity of the CAUR-CQDs/PU and GQD50/PU composite films against three bacterial strains: *P. aeruginosa*, *S. aureus*, and *E. coli*. Figure 7a shows that the GQD50/PU composite films reduce *P. aeruginosa* biofilms by almost 50% compared to the results for the control, with or without blue light irradiation for 1 h.

In Figure 7b, we noticed that these composite films did not show antibiofouling activity. But under blue light irradiation, both composite films reduced *S. aureus* biofilms by almost 85%. The CAUR-CQDs/PU composite films eradicate *E. coli* biofilm completely after 1 h of blue light irradiation, whereas under the same condition, the GQD50/PU composite films reduced the biofilm by about 70%. These results indicate that the antibiofouling activity of the investigated composite films depends on the structure of the nanoparticles and their interaction with biofilms of different types of bacteria.

### 3.8. Biocompatibility Studies

#### 3.8.1. Cytotoxicity

The potential skin toxicity of the investigated composite films was examined using the in vitro model for testing non-tumorigenic human epidermal keratinocyte (HaCaT) cells [44,45]. The cells were treated for 24 h with increasing concentrations (1, 10, 25, 50, 75, and 100%) of all the tested materials (PU, CAUR-CQDs/PU and GQD50/PU), and their viability was measured using an MTT assay. As presented in Figure 8, the observed viability of the HaCaT cells was over 80% for all samples. The obtained results showed that none of the investigated composite films showed any cytotoxic effects on human HaCaT cells, even at a 100% concentration.

#### 3.8.2. Hemolysis Assay

Apart from cytotoxicity, as a part of a biocompatibility’s studies, we investigated the hemolytic properties of the neat PU, CAUR-CQDs/PU, and GQD50/PU composite films. The hemolysis test is used to determine the capacity of the material to induce the destruction of red blood cells and hemoglobin release. As presented in Figure 9, panel a, all tested samples exhibited good blood compatibility, with hemolysis ratios between 0.23% for GQD50/PU and 0.61% for PU (Figure 9, panel a). A visual inspection of the tested composite films samples (Figure 9, panel b) showed that hemoglobin was observed only in the positive control sample, while the color of the other samples varied from clear to pale yellow, without any signs of hemoglobin outflow. The hemolysis ratio for all tested composite films samples did not exceed 1%, which classified them as safe materials, with no hemolytic activity (values < 5% are considered to show a low hemolysis ratio and indicate good blood biocompatibility).

## 4. Discussion

In this study, we investigated the structural, nanoelectrical, and nanomechanical (Young’s modulus of elasticity) properties of two types of carbon based/polyurethane composite films and their effect on their antibacterial, antibiofouling, cytotoxic, and hemolytic properties. The first type of polyurethane composite films was encapsulated by CAUR-CQDs nanoparticles, i.e., N-doped CQDs, and the other type of polyurethane films was encapsulated by GQDs irradiated by gamma rays at a dose of 50 kGy. These composite films differ in particle size (with diameters of 4.2 nm compared to 40 nm) and the types of elements detected by FTIR and XPS, as well as in their characteristic bonds. The encapsulated nanoparticles formed clusters inside the polymer matrix and unevenly distributed within the polymer. The RMS of both composite films differs from that of the neat PU. The surface of the CAUR-CQDs/PU composite films becomes smoother, whereas the surface of the GQD50/PU composite films becomes rougher. We suppose that the difference is due to the particle size of these nanoparticles. This property affects the process of bacterial adhesion on the polymer surface. Both composite films produced singlet oxygen but did not produce hydroxyl radicals. The CAUR-CQDs/PU composite films acted bactericidally against a wide range of tested bacterial strains, whereas the GQD50/PU composite films showed potent antibacterial activity against *E. faecalis*, *K. pneumonie*, and *L. mitocytogenes.* Both composite films were very potent against *E. faecalis*, which was significant due to the fact that this bacterial strain appears frequently in hospitals and causes nosocomial infections [46]. Their antibiofouling activity differed depending on the tested bacteria biofilms. It was interesting to note that the GQD50/PU composite films were more effective against *P. aeruginosa* compared to the CAUR-CQDs/PU composite films, but the *E. coli* biofilms were more sensitive to the latter composite films. The ability to reduce the *P. aeruginosa* biofilm is very important because this bacterium is frequently the cause of nosocomial infections [47]. Thus, it is of utmost interest to apply these types of photodynamic coatings in hospitals to reduce nosocomial infections, particularly in intensive or burns care units. The cytotoxicity and hemolysis tests showed that both composite films were safe materials, without cytotoxic or hemolytic effects.

## 5. Conclusions

In this paper, two types of polyurethane-based composite films were prepared (CAUR-CQDs/PU and GQD50/PU). The CAUR-CQDs nanoparticles were prepared from citric acid and urea using the solvothermal procedure and further encapsulated into polyurethane films. The other sample, GQD50, was prepared using an electrochemical procedure employing graphite electrodes and further irradiated by gamma rays at a dose of 50 kGy. The GQD50 nanoparticles were then encapsulated into polyurethane films. Structural analysis conducted by AFM showed that these dots differed in size, shape, and nanoelectrical and nanomechanical properties. The CAUR-CQDs/PU composite films exhibit bactericidal effects against all tested bacterial strains, except *P. aeruginosa*, *L. monocytogenes*, and *E. coli,* whereas the GQD50/PU composite films predominantly killed *E. faecalis*, *K. pneumonie*, and *L. monocytogenes*. Apart from the other two types of tested bacterial biofilms (*S. aureus* and *E. coli*), the GQD50/PU composite films act bactericidally against *P. aeruginosa* under blue light irradiation for 1 h. Biocompatibility studies showed that these composite films did not exhibit any cytotoxic or hemolytic effects, which indicated that these materials could be used as safe antibacterial surface coatings.

## Figures and Tables

**Figure 1 pharmaceutics-16-01565-f001:**
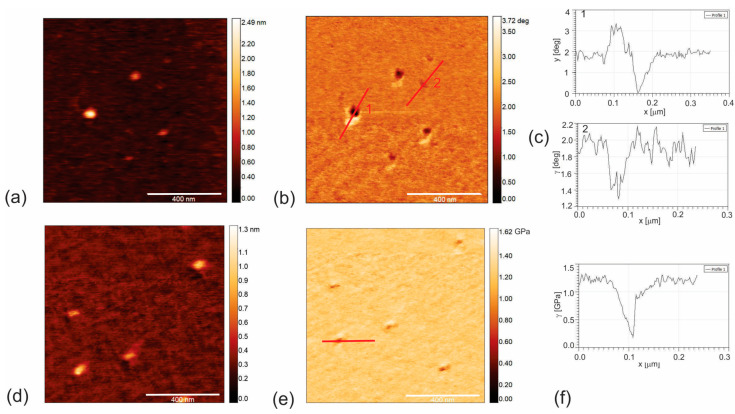
(**a**) Top view AFM image of CAUR-CQDs nanoparticles (height retrace mode) and (**b**) corresponding EFM image (nap retrace mode); (**c**) charge distribution of CAUR-CQDs nanoparticles with larger diameters (number 1) and smaller diameters (number 2); (**d**) top view AFM image (height retrace mode) and (**e**) corresponding image of Young’s modulus of elasticity (Young’s retrace mode) of CAUR-CQDs nanoparticles; (**f**) profile of Young’s modulus of elasticity of CAUR-CQDs.

**Figure 2 pharmaceutics-16-01565-f002:**
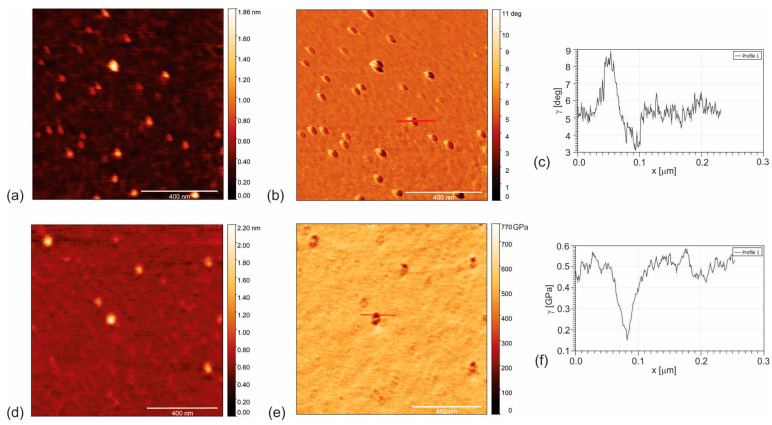
(**a**) Top view AFM image of GQD50 nanoparticles (height retrace mode) and (**b**) corresponding EFM image (nap retrace mode); (**c**) charge distribution of GQD50 nanoparticles; (**d**) top view AFM image (height retrace mode) and (**e**) corresponding image of Young’s modulus of elasticity (Young’s retrace mode) of GQD50 nanoparticles; (**f**) profile of Young’s modulus of elasticity of GQD50 nanoparticles designated by a red line in Figure 2e. The charge distribution of a single GQD50 nanoparticle is presented in Figure 2c. From this figure, we observed that this nanoparticle had two opposite charges: positive and negative.

**Figure 3 pharmaceutics-16-01565-f003:**
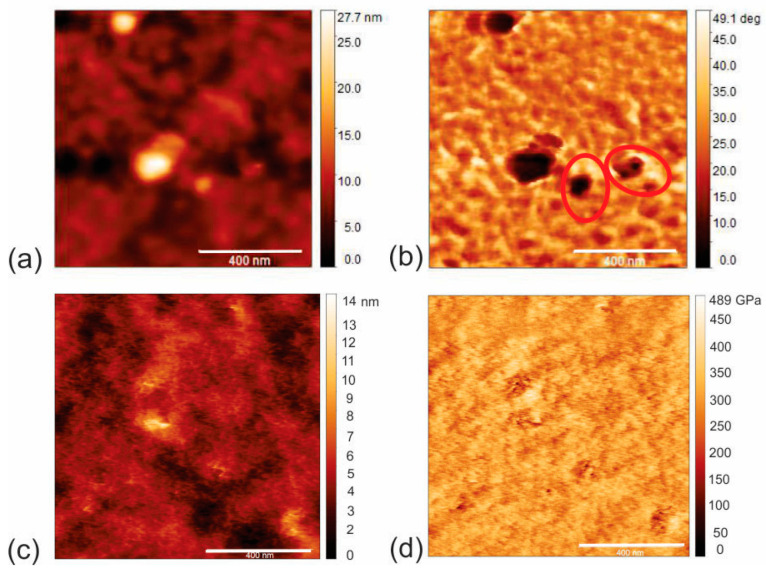
(**a**) AFM image of surface morphology (height retrace), (**b**) EFM image of CAUR-CQDs/PU composite films, (**c**) AFM image of surface morphology (height retrace) and (**d**) corresponding AMFM image of surface of CAUR-CQDs/PU composite films (Young’s retrace).

**Figure 4 pharmaceutics-16-01565-f004:**
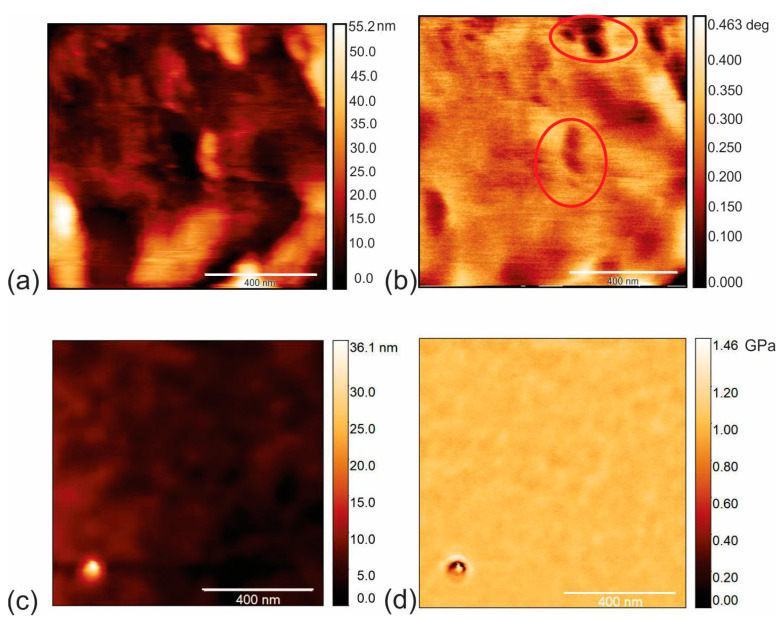
(**a**) AFM image of surface morphology (height retrace), (**b**) EFM image of GQD50/PU composite films, (**c**) AFM image of surface morphology (height retrace) and (**d**) corresponding AMFM image of surface of GQD50/PU composite films (Young’s retrace).

**Figure 5 pharmaceutics-16-01565-f005:**
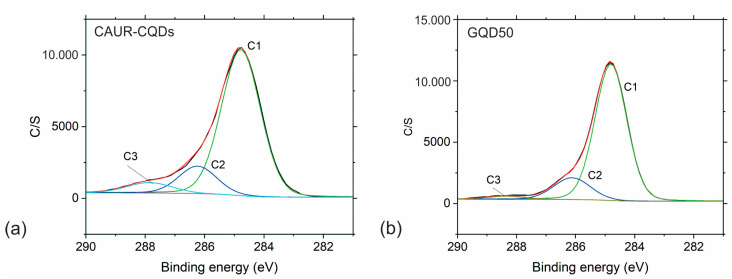
XPS deconvoluted C1s spectra of (**a**) CAUR-CQDs and (**b**) GQD50 nanoparticles.

**Figure 6 pharmaceutics-16-01565-f006:**
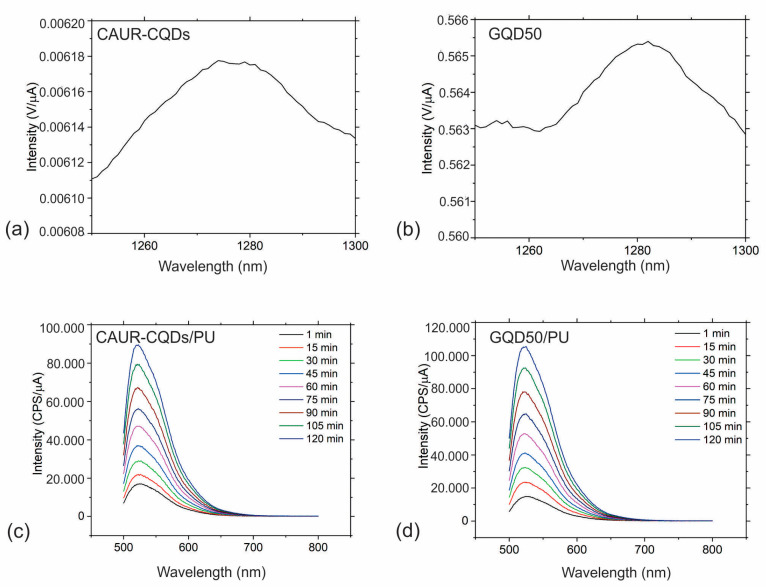
Singlet oxygen generation determined by two methods: the measurement of luminescence of singlet oxygen at 1280 nm of (**a**) CAUR-CQDs nanoparticles and (**b**) GQD50 nanoparticles; the measurement of the PL of SOSG used as a fluorescence probe at 530 nm: (**c**) CAUR-CQDs/PU and (**d**) GQD50/PU composite films samples.

**Figure 7 pharmaceutics-16-01565-f007:**
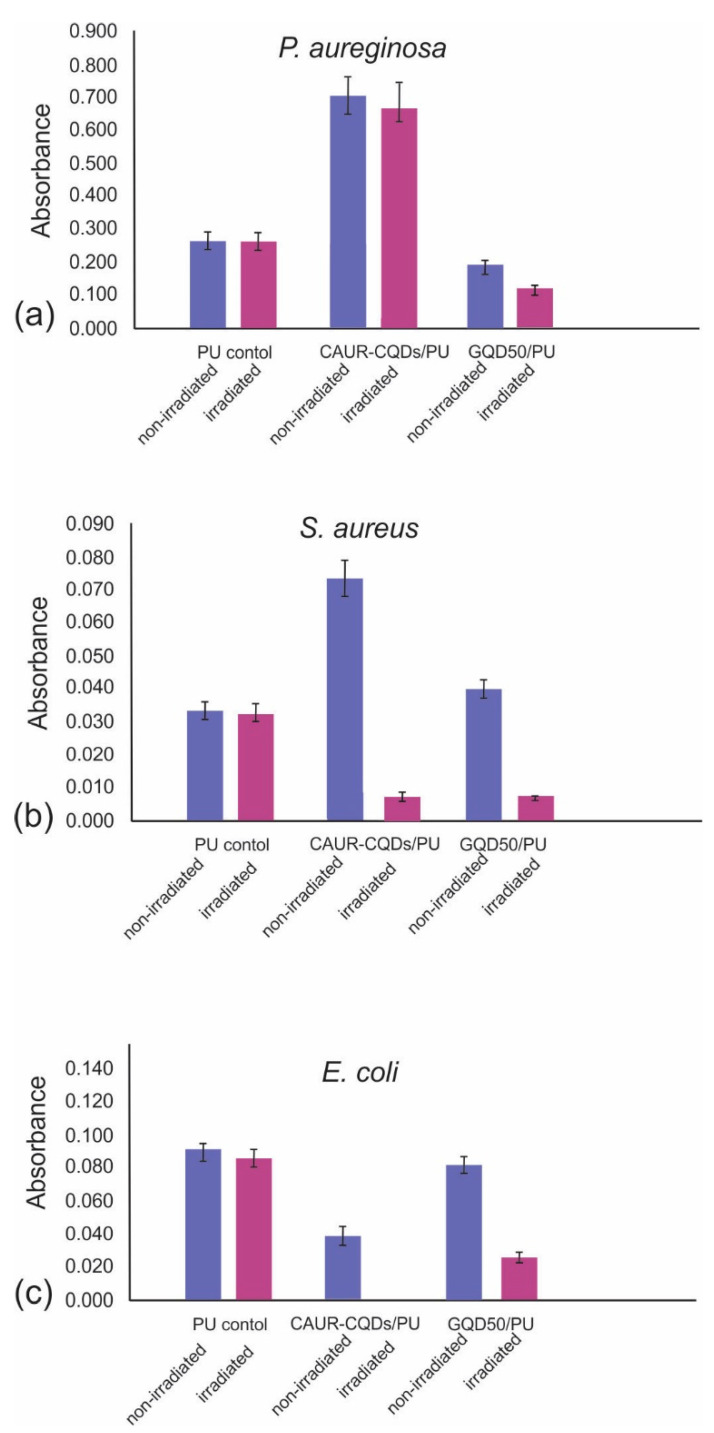
Antibiofouling activity of CAUR-CQDs/PU and GQD50/PU polymer composite films against (**a**) *P. aeruginosa*, (**b**) *S. aureus*, and (**c**) *E. coli* bacterial strains, respectively. As a control, neat PU film was used. All experiments were performed in triplicate.

**Figure 8 pharmaceutics-16-01565-f008:**
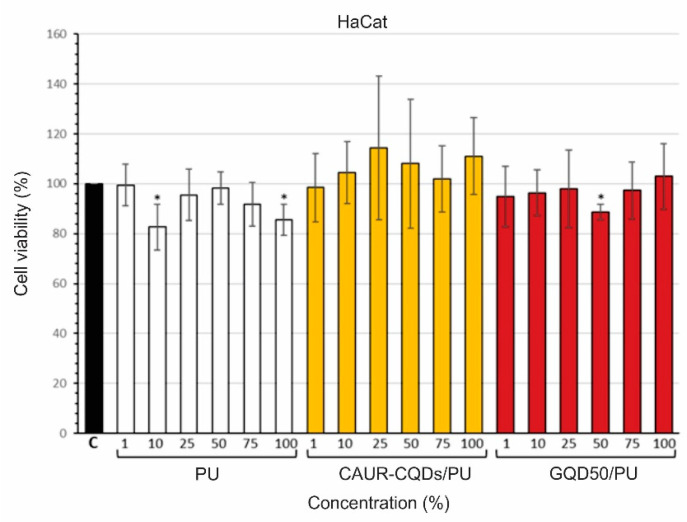
Cytotoxicity effects of PU, CAUR-CQDs/PU, and GQD50/PU toward human epidermal keratinocyte HaCaT cells. Cells were treated for 24 h with the indicated concentrations of the medium (1, 10, 25, 50, 75%, and 100%) previously incubated with the tested materials for 24 h. Values are normalized to untreated control cells (C), expressed as 100%. Data are presented as means ± SD of at least three independent experiments performed in triplicate. Asterisk (*) indicates *p* < 0.05.

**Figure 9 pharmaceutics-16-01565-f009:**
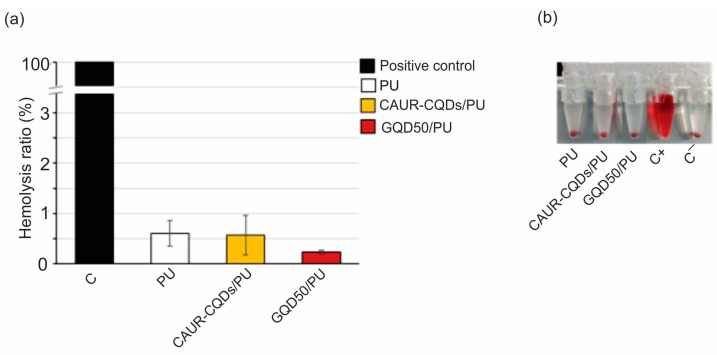
Quantitative and qualitative assessment of hemolytic effect of PU, CAUR-CQDs/PU, and GQD50/PU. (**a**) Hemolytic effect is represented by hemolysis ratio of human RBCs after 1 h of incubation at 37 °C in PBS, previously conditioned by 30 min of incubation with the indicated material. Data are presented as mean ± SD of at least three independent experiments performed in triplicate. C represents the control reaction, i.e., RBC incubated in deionized water. (**b**) Image of pelleted RBC samples after incubation in PBS previously conditioned by incubation with tested materials (PU, CAUR-CQDs/PU, and GQD50/PU), positive control C+ (RBC incubated in deionized water), and negative control C- (RBC incubated in PBS).

**Table 1 pharmaceutics-16-01565-t001:** The elemental composition of CAUR-CQDs and GQD50 samples.

Sample	C (at%)	O (at%)	N (at%)
CAUR-CQDs	83	9.3	7.7
GQD50	83.2	16.4	0.4

**Table 2 pharmaceutics-16-01565-t002:** The characteristic bonds identified in CAUR-CQDs and GQD50 samples.

Sample	C1	C2	C3	O1	O2	O3
CAUR-CQDs	284.8 eV	286.3 eV	288.9 eV	531.1 eV	532.3 eV	533.6 eV
Bond assignment	C-C	C-N	C=O	O=C-N	O=C	O-C
% of bonds	79.3	14.5	6.3	59.8	30.7	9.6
GQD50	284.8	286.1	288.4	531.1	532.4	533.5
Bond assignment	C-C	C–O	C=O/O-C–O	C=O (aromatic)	O=C (aliphatic)	O-C
% of bonds	82.7	15.3	2.1	23.9	58.6	17.6

**Table 3 pharmaceutics-16-01565-t003:** Antibacterial activity R of CAUR-CQDs/PU and GQD50/PU composite films under irradiation with blue light for 1 h.

Bacterial Strains	R_CAUR-CQDs/PU_	R_GQD50/PU_
*S. aureus*	5.2	0.13
*MRSA*	4.3	0.5
*E. faecalis*	4.7	1.12
*P. aeruginosa*	0.06	0.05
*K. pneumonie*	5.3	1.5
*L. monocytogenes*	0.06	1.04
*E. coli*	0.02	0.02
*A. baumanii*	4.9	0.96

## Data Availability

Data are contained within the article and Appendix A.

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
