# Peer review of "Biocompatible Carbon Dots/Polyurethane Composites as Potential Agents for Combating Bacterial Biofilms: N-Doped Carbon Quantum Dots/Polyurethane and Gamma Ray-Modified Graphene Quantum Dots/Polyurethane Composites"

_pharmaceutics, 2024, doi:10.3390/pharmaceutics16121565_

Round 1
Reviewer 1 Report
Comments and Suggestions for Authors
The manuscript of Marković et al. describes the preparation and characterization of two types of polyurethane-based composite film containings with different carbon quantum dots, and their antibacterial, antibiofouling, cytotoxic, and hemolytic properties. Authors found that the CAUR-CQDs/PU composite films had bactericidal effects on all tested bacteria strains except P. aeruginosa, L. monocytogenes and E. coli, while the GQD50/PU film killed E. faecalis, K. pneumonie. The materials were effective against microbial biofilms under blue light irradiation. Both composite films were found to be biocompatible and safe for use as antibacterial surface coatings, without exhibiting any cytotoxic or hemolytic effects.
My main experimental concern with the study (hence the recommendation of major revisions) is the absence of control experiments related to the antibacterial activity of the composite films. With reference to table 3:
1. What do numbers actually mean? What did the authors actually measured? I understand the reference was made to previous studies, but this type of info should be provided here to the reader.
2. What about antimicrobial activity of unmodified PU films under the same irradiation conditions? (= control). An argument is made for carbon quantum dots derived ROS, but the effect of UV light itself should be checked in the controls.
3. How long after inoculation with bacteria were the composite films samples tested/UV-treated?
4. It would be interesting to show the growth of bacteria on the films (including control) over a period of time (e.g. 12h) before UV light treatment. Such eperiment could be carried out under dark and ambient light conditions.
5. Are films effective at preventing bacterial growth without UV-irradiation? I ask this Q in terms of possible applications of the films. If the same are intended to coat e.g. small devices, a short UV sterilizing irradiation makes sense and it would be technically feasible; but what about larger surfaces?
General comments: please avoid single sentence paragraphs (e.g. lines 60-62 and 508-510). Indeed, e.g. 55-62 can be grouped in a single paragraph without loss of meaning.
Comments on the Quality of English LanguageOverall the manuscript is fairly well-written (I had no problems following the author's descriptions and arguments), however small grammatical mistakes are scattered throughout the text. Nothing serious, but ENG editing is required.
Author Response
Dear Sir,
We corrected manuscript as referees requested. All changes in the manuscript have been highlighted. The answers to referees are below.
- My main experimental concern with the study (hence the recommendation of major revisions) is the absence of control experiments related to the antibacterial activity of the composite films. With reference to table 3:
All antibacterial tests were performed according to ISO standard ISO 22196 (Plastics – Measurement of antibacterial activity on plastic surfaces); The antibacterial activity was determined by the following way:
N- Number of viable bacteria recovered for test specimen
N=(100xCxDxV)/A (1)
C-average plate count, D-dilution factor for the plate counted, V-volume (mL) of SCDLP added, A-surface area (mm2) of the cover film
R -Antibacterial activity of tested material
R=(Ut-U0)-(At-U0)=Ut-Аt (2)
U0 – average of log10 of the number of bacteria from non-irradiated sample in 0 h, Ut – average log 10 of the number of bacteria from non-irradiated sample after 24 h, At – average log 10 of the irradiated sample after 24 h.
We inserted complete procedure in the text of the manuscript (section 2.4).
- What do numbers actually mean? What did the authors actually measured? I understand the reference was made to previous studies, but this type of info should be provided here to the reader.
The antibacterial activity was determined by the following way:
N- Number of viable bacteria recovered for test specimen
N=(100xCxDxV)/A (1)
C-average plate count, D-dilution factor for the plate counted, V-volume (mL) of SCDLP added, A-surface area (mm2) of the cover film
R -Antibacterial activity of tested material
R=(Ut-U0)-(At-U0)=Ut-Аt (2)
U0 – average of log10 of the number of bacteria from non-irradiated sample in 0 h, Ut – average log 10 of the number of bacteria from non-irradiated sample after 24 h, At – average log 10 of the irradiated sample after 24 h.
We inserted complete procedure in the section 2.4.
- What about antimicrobial activity of unmodified PU films under the same irradiation conditions? (= control). An argument is made for carbon quantum dots derived ROS, but the effect of UV light itself should be checked in the controls.
The experiments were performed on all specimens: neat PU, CAUR-CQDs/PU, GQD50/PU (non-irradiated) and CAUR-CQDs/PU, GQD50/PU (irradiated by blue light at 470 nm for 1 h). Neat PU did not show any antibacterial activity (expected), non-irradiated specimens with carbon dots encapsulated showed very weak antibacterial activity. We did not use UV light for experiments because scientific reports which indicated that under 405 nm wavelengths, the light killed bacteria strains itself (ref. M.D. Barneck, N.L.R. Rhodes, M. De la Presa, J.P. Allen, A.E. Poursaid, M.M. Nourian, M.A. Firpo, J.T. Langell, Violet 405-nm light: a novel therapeutic agent against common pathogenic bacteria, J. Surg. Res. 206 (2016) 316–324, https://doi.org/10.1016/j.jss.2016.08.006.)
- How long after inoculation with bacteria were the composite films samples tested/UV-treated?
All tests were performed in triplicate, and adequate controls for non-irradiated samples were used (three to measure viable cells immediately and three to measure viable cells after incubation for 24 h). All tested materials-CAUR-CQDs/PU and GQD50/PU composite films were prepared as flat squares, 25 mm × 25 mm, sterilized by UV lamp at 258 nm for 30 min. Non-irradiated specimens were used for control of bacterial viability, and others materials specimens were incubated under blue light at 470 nm for 1 h. The composite films samples were inoculated with 0.2 mL bacteria suspension of 1−5×108 cell/mL. Test inoculums were covered with films, and petri dishes with test specimens were incubated for 24 h at 35 °C, humidity 90 %. Bacteria were recovered with 10 mL of SCDLP, and 10 µL (different dilutions) were plated on LB agar and incubated 24 h at 35 oC, after which bacterial colonies were counted. The distance from the blue lamp and samples was 50 cm to provide homogeneous irradiation of the samples.
- It would be interesting to show the growth of bacteria on the films (including control) over a period of time (e.g. 12h) before UV light treatment. Such experiment could be carried out under dark and ambient light conditions.
We did not use UV irradiation of composite films. We used visible light at 470 nm (blue light) because light at 405 nm and lower wavelengths killed bacteria itself. These composite films become sterile under visible blue light irradiation for 1 h. Tables S1, S2, S3 (supporting information) and Table 3 show all data obtained in the antibacterial testing.
- Are films effective at preventing bacterial growth without UV-irradiation? I ask this Q in terms of possible applications of the films. If the same are intended to coat e.g. small devices, a short UV sterilizing irradiation makes sense and it would be technically feasible; but what about larger surfaces?
We did not use UV irradiation of composite films. We used visible light at 470 nm (blue light) because light at 405 nm and lower wavelengths killed bacteria itself. These composite films become sterile under visible blue light irradiation for 1 h. We produced different types of these coatings (antibacterial door handles, antibacterial stickers, antibacterial cover for laptop keyboards and antibacterial phone case). All these innovations you can see on our Instagram profiles: www.instagram.com/photogun4microbes and www.instagram.com/lumi4cell. We submitted to Serbian paten office patent application under title: Photodynamic antibacterial films for application on highly touched objects.
General comments: please avoid single sentence paragraphs (e.g. lines 60-62 and 508-510). Indeed, e.g. 55-62 can be grouped in a single paragraph without loss of meaning.
We corrected the manuscript as requested.
Comments on the Quality of English Language
Overall, the manuscript is fairly well-written (I had no problems following the author's descriptions and arguments), however small grammatical mistakes are scattered throughout the text. Nothing serious, but ENG editing is required.
We corrected grammatical and typo errors in the manuscript.
Best regards,
Zoran Markovic
Reviewer 2 Report
Comments and Suggestions for Authors
quantum dots and composite film in terms of morphology, ROS generation, antimicrobial, anti-biofouling activity, cytotoxicity, and biocompatibility, respectively, and it is recommended to accept it with modifications.
1.Two kinds of carbon quantum dots are studied in this work, please explain why these two materials are studied in the article;
2.There is a lack of comparison between the two types of carbon quantum dots in the article, please make a corresponding comparison and analysis;
3.Biocompatibility includes more than hemolysis rate test, and it is suggested to combine cytotoxicity into the biocompatibility part;
Please add the relationship between anti-biofouling activity and antimicrobial activity in detail;
4.The characterization part of the whole paper lacks the discussion of the connection between each other, and it is suggested to integrate the whole paper with the discussion of antimicrobial activity, i.e., combine the discussion with the characterization part, and do not take out the discussion separately;
5.It is suggested to touch up the diagrams, including the text, clarity and overall aesthetics of the diagrams;
6.In“3.2. Surface morphology of CAUR-CQDs/PU and GQD50/PU composite films”,the author compared the Root-mean-square (RMS) roughness of the polymer composite with thee value of RMS of neat PU film,but the analysis of this data is lacking.
7.No analysis of nanoparticle distribution uniformity was found.The whole study is done in a comprehensive way, characterization and analysis of carbon.
Comments on the Quality of English LanguageThere are many grammatical errors in the text, which need to be further refined.
Author Response
Dear Sir,
We corrected manuscript as referees requested. All changes in the manuscript have been highlighted. The answers to referees are below.
quantum dots and composite film in terms of morphology, ROS generation, antimicrobial, anti-biofouling activity, cytotoxicity, and biocompatibility, respectively, and it is recommended to accept it with modifications.
Dear Sir, Thank you for the recommendation.
1.Two kinds of carbon quantum dots are studied in this work, please explain why these two materials are studied in the article;
In our investigation we chose to investigate two different carbon-based nanoparticles encapsulated in the polyurethane because these nanoparticles have different properties: the first one CAUR-CQDs were synthesized by bottom-up method and the others were synthesized by top down method (section 2.2.1). Nanoparticles produced by different procedures and with different starting materials would have different properties (different carbon core structure and different functional groups distributed over the surface and edges of carbon basal plane). Their different structural properties would affect their ability to produce reactive oxygen species and further their antibacterial activities.
2.There is a lack of comparison between the two types of carbon quantum dots in the article, please make a corresponding comparison and analysis;
-We compared their structure (EFM and AMFM images (Figures 1 and 2) and established the differences between them. We compared their antibacterial and antibiofouling activities (sections 3.6 and 3.7).
3.Biocompatibility includes more than hemolysis rate test, and it is suggested to combine cytotoxicity into the biocompatibility part;
-We corrected manuscript as requested and put cytotoxicity and hemolysis studies in one section Biocompatibility studies (section 3.8).
Please add the relationship between anti-biofouling activity and antimicrobial activity in detail;
-There is no relation between antibacterial and antibiofouling activities. It is two different types of tests. With first one we investigated the ability of composite films to kill different types of planktonic bacteria strains. The second test (antibiofouling test) shows the ability of composite films to reduce or even eliminate completely bacteria biofilms. Reduction or elimination of bacteria biofilms formed on various surfaces is very hard process because bacteria formed polysaccharide matrix.
4.The characterization part of the whole paper lacks the discussion of the connection between each other, and it is suggested to integrate the whole paper with the discussion of antimicrobial activity, i.e., combine the discussion with the characterization part, and do not take out the discussion separately;
-In our previous papers we got advice from referees to put always discussion separately due to easier reading of manuscript.
5.It is suggested to touch up the diagrams, including the text, clarity and overall aesthetics of the diagrams;
-We corrected manuscript as requested.
6.In“3.2. Surface morphology of CAUR-CQDs/PU and GQD50/PU composite films”,the author compared the Root-mean-square (RMS) roughness of the polymer composite with thee value of RMS of neat PU film,but the analysis of this data is lacking.
-We put the discussion about RMS data (section 3.2).
7.No analysis of nanoparticle distribution uniformity was found. The whole study is done in a comprehensive way, characterization and analysis of carbon.
-We corrected manuscript as requested (section 3.2).
Comments on the Quality of English Language
There are many grammatical errors in the text, which need to be further refined.
-We corrected manuscript as requested.
Best regards,
Zoran Markovic
Round 2
Reviewer 1 Report
Comments and Suggestions for Authors
Overall, authors have clarified and answered my questions, thus I can recommend to accept the paper.
Comments on the Quality of English LanguageEnglish is fine. Just editorial work required.